# Preterm birth and secondhand smoking during pregnancy: A case–control study from Vietnam

Nguyen N. Rang[1,2]*, Tran Q. Hien[3], Ton Q. Chanh[2], Tran K. Thuyen[3]

1 Department of Pediatrics, Can Tho University of Medicine and Pharmacy, Can Tho, Vietnam,
2 Department of Pediatrics, Women and Children Hospital of An Giang, Long Xuyen, An Giang, Vietnam,
3 Department of Obstetrics and Gynecology, Women and Children Hospital of An Giang, Long Xuyen, An Giang, Vietnam

* nguyenngocrang@gmail.com

**Data Availability Statement:** All relevant data are within the manuscript and its Supporting Information files.

**Funding:** The authors received no specific funding for this work.

## Abstract

### Background

The relationship between women who are exposed to secondhand smoke and preterm birth is still controversial. The present study aimed to examine the association between maternal secondhand smoking (SHS) during pregnancy and preterm birth.

### Methods

A 1:1 case-control study was conducted at delivery room of The Women's and Children's Hospital of An Giang, Vietnam. A total of 288 cases of preterm birth and 288 controls included in this study. A structured questionnaire in a face-to-face interview was used to assess SHS and potential confounders (maternal age, body mass index, occupation, education level, parity, antenatal care visits, history of preterm birth, prenatal bleeding and pre-eclampsia/eclampsia).

### Results

SHS was reported more frequently by women who delivered preterm babies compared with women of term deliveries (67.4% vs. 51.0%; P <0.001). After controlling all potential confounders, multivariable logistic regression analysis showed a relationship between SHS during pregnancy and preterm delivery (adjusted Odds ratio: 1.92; 95% CI 1.31, 2.81)

### Conclusions

Our findings suggest that exposure to household tobacco smoke during pregnancy is associated with preterm birth.

**Competing interests:** The authors have declared that no competing interests exist.

## Introduction

Preterm birth (PTB) is defined as a delivery that occurs before 37 weeks of pregnancy. Preterm birth is the leading cause of perinatal morbidity and mortality in both developed and developing countries [1].

Epidemiological studies have suggested that sociodemographic and pregnancy specific factors may increase the risk of preterm birth. In addition, alcohol and cigarette use have been associated with the risk of preterm delivery. However, there are still many cases with unknown etiology [2].

There has been evidence that women who smoke cigarettes are at risk of preterm birth [3], but the relationship between women who are exposed to secondhand smoke and preterm birth was still controversial.

In Vietnam, the prevalence of smoking among women was very low (1.1%), but the prevalence of smoking among men was high (45.3%) [4] and the exposure to secondhand smoke at home accounted for 73.1% [5]. A recent systematic review suggested a possible relationship between maternal secondhand smoking (SHS) and preterm birth [6].

We hypothesized that even if a women did not smoke but exposed to cigarette smoke at home from her husband or other family members during pregnancy, she would have a risk of preterm birth.

The aim of this study was to examine the association between maternal secondhand smoking (SHS) during pregnancy and preterm birth.

## Materials and methods

Design: It is a 1:1 matched case-control study. Cases of preterm birth were singleton babies born before the 37th gestational weeks. For every premature delivery, one full-term consecutive delivery was taken as a control. Women with active smoking or women having babies with congenital abnormalities or still births were excluded from the study.

Setting: Cases and controls were recruited between June 2018 and June 2019 at delivery rooms of The Women and Children Hospital of An Giang, Vietnam.

Sample size: We calculated sample size for matched case control (1:1 case to control ratio) by assuming odds ratio of 1.99 [7]. With significance level set to 0.05, power to 0.80, the calculated sample size was 288 matched pairs.

### Instrument and measurement

The face-to-face interview was used to collect data from the respondents, The questionnaire consisted of 3 components, namely social demography (age, height, weight, occupation, educational status), history of obstetrics (number of antenatal care visits, parity, history of preterm births, vaginal bleeding before or after 28 weeks of gestation, and preeclampsia/ eclampsia during pregnancy), and history of secondhand smoking. The questionnaire was first developed in English and was translated into Vietnamese. The questionnaire was also pre-tested in 40 respondents in a pilot study at the Obstetrics ward of Women and Children Hospital of An Giang. Minor corrections were made on the questionnaire according to the outcomes and feedback in the pre-testing study. The secondhand smoking questionnaire is presented in the Supporting information (**S1 Questionnaire**).

**Definition of secondhand smoke exposure.** Pregnant women who did not smoke but were exposed to husband or family members who smoked on a daily basis of more than 5 cigarettes a day at home (exposure to environmental tobacco smoke was not included) were considered as exposed to SHS.

**Definition of preterm birth.** Preterm birth: babies born before the 37th gestational weeks. The gestational age was determined by ultrasound examination or based on the first day of the last menstrual period.

## Definition of confounding factors

Education level: categorized as primary and secondary school

Occupation: categorized as farmer and non-farmer (housewife, government worker, merchant)

History of preterm birth: categorized as Yes or No

Parity was reported as the number of previous pregnancies lasting more than 22 weeks gestation.

Antenatal care visits: defined as non-adequacy < four times or adequacy ≥ four times during pregnancy.

Prenatal bleeding: any vaginal bleeding before or after 28 weeks of gestation during pregnancy and categorized as Yes or No

Preeclampsia or eclampsia: categorized as Yes or No

Body mass index (BMI) was calculated as weight (in kilograms) divided by the square of height (in meters).

## Ethical consideration

The research protocol was executed according to the Helsinki Declaration with the approval of the Ethics Committee of The Women and Children of An Giang (Approval No. 32a-QĐBVSN). Written consent was obtained from each respondent prior to the interview by trained research assistants.

## Statistical analysis

Categorical variables were expressed as numbers and frequencies (%).Chi-square test or with Fisher exact test when appropriate were used to testing relationships between categorical variables.

Multivariate logistic regression model was used to calculate odds ratio (OR) and its 95% confidence intervals (CIs) for the association between SHS and preterm birth. In this analysis, all potential confounders (maternal age, BMI, maternal occupation, education level, parity, ANC visits, history of preterm birth, prenatal bleeding, preeclampsia/eclampsia) were included.

Logistic regression with Enter method was used and the Hosmer and Lemeshow goodness-of-fit test was used to determine if the model fit the data. Confidence intervals, at the 95% level were also reported for each adjusted OR.

The level of significance was $p < 0.05$. The SPSS Statistics for Windows version 22.0 was used for all statistical analyses.

## Results

A total of 288 cases of preterm birth and 288 controls included in this study. SHS was reported more frequently by women who delivered preterm babies compared with women of term deliveries 67.4% (194/288) vs. 51.0% (147/288) ($P < 0.001$).

The sociodemographic data, reproductive and pregnancy-related characteristics among cases and controls are shown in Table 1.

**Table 1. Association between socio-demographic, reprodutive and pregnancy-related factors and preterm birth.**

| Characteristics | Full-term (n = 288) | | Preterm (n = 288) | | P value |
|---|---|---|---|---|---|
| | No. | % | No | % | |
| Age, years | | | | | |
| ≤ 20 | 31 | 10.8 | 33 | 11.5 | |
| 21–34 | 228 | 79.2 | 215 | 74.7 | > 0.05 |
| ≥35 | 29 | 10.0 | 40 | 13.8 | |
| Weight, kg | | | | | |
| ≤45 | 108 | 37.5 | 113 | 39.2 | |
| 46–54 | 145 | 50.3 | 121 | 42.0 | < 0.05 |
| ≥55 | 35 | 12.2 | 54 | 18.8 | |
| Height, cm | | | | | |
| ≤ 145 | 15 | 5.2 | 15 | 5.2 | |
| 146–154 | 132 | 45.8 | 131 | 45.5 | > 0.05 |
| ≥155 | 141 | 49.0 | 142 | 49.3 | |
| BMI, % | | | | | |
| ≤ 18.5 | 69 | 24.0 | 65 | 22.6 | |
| 18.6–24.9 | 212 | 73.6 | 201 | 69.8 | < 0.05 |
| ≥25 | 7 | 2.4 | 22 | 7.6 | |
| Education, years | | | | | |
| Primary (0–5) | 46 | 16.0 | 69 | 24.0 | < 0.05 |
| High school (≥6) | 242 | 84.0 | 219 | 76.0 | |
| Occupation | | | | | |
| Farmer | 68 | 23.6 | 89 | 30.9 | < 0.05 |
| Non-farmer | 220 | 76.4 | 199 | 69.1 | |
| Parity | | | | | |
| Nulliparous | 152 | 52.8 | 163 | 56.6 | > 0.05 |
| Parous (≥2) | 136 | 47.2 | 125 | 43.4 | |
| ANC visit (numbers) | | | | | |
| 0–3 | 81 | 28.1 | 118 | 41.0 | < 0.001 |
| ≥ 4 | 207 | 71.9 | 170 | 59.0 | |
| Previous preterm births | | | | | |
| No | 282 | 97.9 | 249 | 86.5 | < 0.001 |
| Yes | 6 | 2.1 | 39 | 13.5 | |
| Prenatal bleeding | | | | | |
| No | 286 | 99.3 | 255 | 88.5 | < 0.001 |
| Yes | 2 | 0.7 | 33 | 11.5 | |
| Pre-eclampsia/Eclampsia | | | | | |
| No | 285 | 99.0 | 256 | 88.9 | < 0.001 |
| Yes | 3 | 1.0 | 32 | 11.1 | |
| Secondhand smoking | | | | | |
| No | 141 | 49.0 | 94 | 32.6 | < 0.001 |
| Yes | 147 | 51.0 | 194 | 67.4 | |

ANC, Antenatal care

The maternal age, the height and parity were not different between cases and controls. Compared with women delivering full-term babies, women delivering preterm babies had higher weight and BMI. Higher proportion of the women of the cases were farmers and had

fewer years of education than that of the controls. The number of ANC visits were fewer in women who delivered preterm babies. Women who had history of preterm delivery, prenatal bleeding and pre-eclampsia/eclampsia during pregnancy had higher proportion of preterm birth.

After controlling all potential confounders (maternal age, BMI, maternal occupation, education level, parity, ANC visits, history of preterm birth, prenatal bleeding, preeclampsia/eclampsia) in a multivariate analysis model, the study showed that women with SHS were high risk of preterm birth as compared to women without SHS during pregnancy. The adjusted odds ratio (AOR) for SHS was 1 .92 (CI 95%:1.31–2.81) (Table 2).

## Discussion

In this case control study, after controlling all confounders related to socio-demographic and reproductive characteristics, we found that non-smoking women with exposure to secondhand smoke from the household had an increased risk of having a preterm birth. To the best of our knowledge, this paper is the first to study the association between SHS and preterm birth in Vietnam. High levels of tobacco exposure at home may support our positive results as SHS causes preterm birth in Vietnam where 45.3% of men currently smoked cigarette and 73.1% of non-smokers were exposed to tobacco smoke at home [4, 5].

Our research results were also consistent with other studies in Asian countries where the level of tobacco exposure at home and the environment was high [8–12]. However, the study of Qiu et al. [12] in China only found an association between SHS and infants of very preterm birth (<32 weeks of gestational age).

Our results were in line with one recent study in Texas, USA. Hoyt et al. [7] reported that women with household and workplace/school SHS were increased the risk for preterm birth (AOR 1.99; 95% CI 1.13–3.50), whereas the majority of previous studies in developed countries have found weak or no association between SHS and premature birth [13–23]. A meta-analysis of 58 trials showed that passive tobacco smoking increased the incidence of low birth weight infants by 22%; however, it was unclear whether it caused preterm birth [18]. The evidence of the association between passive cigarette exposure and preterm birth is unclear in developed countries. This can be explained by lower levels of tobacco smoke pollution in the environment and at home. In fact, the number of smokers in developed countries has declined significantly in recent decades and the smoking habits at home were also reduced [24, 25]. Some

**Table 2. Association between secondhand smoking and preterm birth before 37 weeks in multivariable logistic regression analysis.**

| Variables | AOR (95%CI) | P value |
|---|---|---|
| Maternal age | 1.04 (1.00–1.08) | < 0.01 |
| BMI | 1.04 (0.99–1.10) | > 0.05 |
| Occupation (farmer) | 1.03 (0.66–1.61) | > 0.05 |
| Education (primary school) | 1.42 (0.88–2.29) | > 0.05 |
| Parity (parous) | 0.48 (0.30–0.76) | < 0.01 |
| ANC visits (≥4) | 0.53 (0.35–0.79) | < 0.01 |
| History of preterm birth | 10.4 (4.1–26.3) | < 0.001 |
| Prenatal bleeding | 22.6 (5.2–97.7) | < 0.001 |
| Preeclampsia/Eclampsia | 9.55 (2.80–32.4) | < 0.001 |
| Secondhand smoking | **1.92 (1.31–2.81)** | < 0.01 |

AOR, adjusted Odds Ratio; BMI, Body mass index; ANC, Antenatal care

studies suggest that only women exposed to passive tobacco with high doses and long duration had the risk of preterm birth [26, 27].

That SHS affected preterm birth in some developed countries was evidenced by a reduction in rates of preterm birth when smoke-free legislation was implemented [28–32].

The strengths of the current study were that pregnancy-related factors such as prenatal bleeding, preeclampsia were controlled as compared to previous studies and most of women (1,1%) are non-smokers, thus eliminating the confounders from active smoking.

The limitations of this study were: First, maternal SHS exposure was estimated through face-to-face interview; it was not possible to measure nicotine levels in maternal hair or to measure cotinine in urine to determine tobacco exposure. Second, this is a case-control study based on hospitalized patients which often do not represent the general population. Third, due to the retrospective nature of case-control studies, the questionnaires were administered after delivery and therefore bias can not be excluded. Fourth, it is impossible to exclude the possibility of a direct effect of paternal sperm changes from smoking on the fetus. Finally, we cannot collect all possible confounders for preterm birth.

In summary, exposure to household tobacco smoke during pregnancy is associated with preterm birth even after adjusting for all possible confounders. More cohort studies in the future are needed to confirm this finding. Evaluation of tobacco exposure and steps to avoid it during pregnancy should be an important part of antenatal care.

## Supporting information

**S1 Questionnaire.**
(DOCX)

**S1 Dataset.**
(SAV)

## Acknowledgments

We thank the midwife study group in delivery rooms of The Women and Children Hospital of An Giang for their assistance with recruitment and data collection. We also thank the families who participated in this study for contributing their time and expertise.

## Author Contributions

**Conceptualization:** Nguyen N. Rang.

**Data curation:** Tran Q. Hien, Ton Q. Chanh.

**Formal analysis:** Nguyen N. Rang.

**Investigation:** Tran K. Thuyen.

**Supervision:** Tran Q. Hien.

**Validation:** Ton Q. Chanh.

**Writing – original draft:** Nguyen N. Rang.

**Writing – review & editing:** Nguyen N. Rang.

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
