## [Decision Letter · Decision Letter 0]

24 Jun 2020

PONE-D-20-09934

Preterm birth and secondhand smoking during pregnancy: a case–control study from Vietnam

PLOS ONE

Dear Dr. Nguyen,

Thank you for submitting your manuscript to PLOS ONE. After careful consideration, we feel that it has merit but does not fully meet PLOS ONE’s publication criteria as it currently stands. Therefore, we invite you to submit a revised version of the manuscript that addresses the points raised during the review process.

An expert reviewer has reviewed the manuscript. While the work is thought to be of interest, there is concern that that the analytical methods used and current data do not support the conclusions. To address some of the concerns may require additional studies and analyses.

We look forward to receiving your revised manuscript.

Kind regards,

Cheryl S. Rosenfeld, DVM, PhD

Academic Editor

PLOS ONE

Journal Requirements:

2. Please address the following:

- Please include additional information regarding the survey or questionnaire used in the study and ensure that you have provided sufficient details that others could replicate the analyses. For instance, if you developed a questionnaire as part of this study and it is not under a copyright more restrictive than CC-BY, please include a copy, in both the original language and English, as Supporting Information. Please include further details of the development and validation of this questionnaire, including any pre-testing that took place.

- Please refrain from stating p values as 0.000, either report the exact value or employ the format p<0.001.

- Please provide additional details regarding participant consent. In the ethics statement in the Methods and online submission information, please ensure that you have specified what type of consent you obtained (for instance, written or verbal, and if verbal, how it was documented and witnessed).

4. Please include your tables as part of your main manuscript and remove the individual files. Please note that supplementary tables (should remain/ be uploaded) as separate "supporting information" files.

Reviewers' comments:

Reviewer's Responses to Questions

**Comments to the Author**

1. Is the manuscript technically sound, and do the data support the conclusions?

Reviewer #1: Partly

2. Has the statistical analysis been performed appropriately and rigorously? 

Reviewer #1: Yes

3. Have the authors made all data underlying the findings in their manuscript fully available?

Reviewer #1: Yes

4. Is the manuscript presented in an intelligible fashion and written in standard English?

Reviewer #1: Yes

5. Review Comments to the Author

Reviewer #1: This manuscript by Rang et al addresses an important theme of the effect of secondhand smoking on preterm delivery. There have been several comprehensive articles addressing a parallel theme that reported higher incidence of preterm birth in women exposed to secondhand smoking. The authors of this manuscript confirm earlier findings and suggest that this happens to be true in Vietnamese population. The study uses a sizeable cohort but lacks some rigorous protocols that should have been used and described to strengthen the study.

Major comments:

1. The authors should use some of the rigorous tools that have been described in a similar study by Ashford et al (The effects of prenatal secondhand smoke exposure on preterm birth and neonatal outcomes. J Obstet Neonatal Nurs. 2010; 39:525-535).

2. How did the authors confirm exposure to secondhand smoking? Was it solely based on questionnaire or was a maternal hair nicotine estimation performed on subjects and controls? Such a statement or description is not provided. Without such a quantification of maternal hair nicotine, it is hard to base the authors’ conclusions on verbal or written answers.

3. Were the pregnant women asked about the exposure to smoking prior to or after delivery? It is possible that the women looked for a reason for their preterm deliveries if they were asked after delivery.

4. Did the authors compare the incidence of preterm birth between women with active smoking and women with secondhand exposure? This is very important because this has not been studied.

5. Results: If the reviewer understands it clearly, 67.4% women with preterm birth revealed that they were exposed to secondhand smoke compared to 51% with term pregnancy. This means 51% women with term pregnancy also were exposed. Since 51% exposed women experienced term pregnancy, there must be a second confounding factor in women with preterm birth. Can the authors elaborate on this issue?

6. PLOS authors have the option to publish the peer review history of their article (what does this mean?). If published, this will include your full peer review and any attached files.

Reviewer #1: No

---

## [Author Response · Author response to Decision Letter 0]

13 Jul 2020

Response to Reviewer’s

it’s done in Revised Manuscript 

2. Please address the following:

- Please include additional information regarding the survey or questionnaire used in the study and ensure that you have provided sufficient details that others could replicate the analyses. For instance, if you developed a questionnaire as part of this study and it is not under a copyright more restrictive than CC-BY, please include a copy, in both the original language and English, as Supporting Information. Please include further details of the development and validation of this questionnaire, including any pre-testing that took place.

- Please refrain from stating p values as 0.000, either report the exact value or employ the format p<0.001.

- Please provide additional details regarding participant consent. In the ethics statement in the Methods and online submission information, please ensure that you have specified what type of consent you obtained (for instance, written or verbal, and if verbal, how it was documented and witnessed).

The questionnaires in original language (Vietnamese) and English are available in the Supporting Information (S1 questionnaire.docx) 

We included the details of the development and validation of the questionnaires in the Methods of Revised manuscript

We replaced p values as 0.000 with the format <0.001

We provided more details regarding participant consent and signed before

attending the review in Ethical issues of the Revised manuscript 

a. Please clarify the sources of funding (financial or material support) for your study. List the grants or organizations that supported your study, including funding received from your institution.

d. If you did not receive any funding for this study, please state: “The authors received no specific funding for this work.”

We declared “The authors received no specific funding for this work.” in cover letter and in the Revised manuscript

4. Please include your tables as part of your main manuscript and remove the individual files. Please note that supplementary tables (should remain/ be uploaded) as separate "supporting information" files.

 We included the tables in the revised manuscript

Comments to the Author

1. Is the manuscript technically sound, and do the data support the conclusions?

Reviewer #1: Partly

2. Has the statistical analysis been performed appropriately and rigorously?

Reviewer #1: Yes

3. Have the authors made all data underlying the findings in their manuscript fully available?

The dataset of this study is available in the Supporting Information (S2 dataset.sav) 

Reviewer #1: Yes

4. Is the manuscript presented in an intelligible fashion and written in standard English?

Reviewer #1: Yes

5. Review Comments to the Author

Reviewer #1: This manuscript by Rang et al addresses an important theme of the effect of secondhand smoking on preterm delivery. There have been several comprehensive articles addressing a parallel theme that reported higher incidence of preterm birth in women exposed to secondhand smoking. The authors of this manuscript confirm earlier findings and suggest that this happens to be true in Vietnamese population. The study uses a sizeable cohort but lacks some rigorous protocols that should have been used and described to strengthen the study.

Major comments:

1. The authors should use some of the rigorous tools that have been described in a similar study by Ashford et al (The effects of prenatal secondhand smoke exposure on preterm birth and neonatal outcomes. J Obstet Neonatal Nurs. 2010; 39:525-535)..

Measuring nicotine in maternal hair to confirm exposure to secondhand smoking as the study of Ashford et al. may be more reliable than interviewing tobacco exposures by members smoking at home; however, this test is not available in Vietnam due to the cost. According to Perez-Rios M et al. (2013), 84% (309/369) of the secondhand smoking-related studies have used interview questionnaires so far; only 16 % (50/369) of the studies measured cotinine/ nicotine in hair, urine or saliva.

The advantage of questionnaires is that they allow for a detailed ascertainment of exposure and this should be specially valued. Questionnaires also obtain information regarding the intensity and duration of the smoke exposure at home.

In the future, we will design a study to compare the nicotine concentration in hair with the number of cigarettes per day smoked at home. 

Pérez-Ríos M et al. Questionnaire-based second-hand smoke assessment in adults. Eur J Public Health. 2013;23(5):763-7.

 2. How did the authors confirm exposure to secondhand smoking? Was it solely based on questionnaire or was a maternal hair nicotine estimation performed on subjects and controls? Such a statement or description is not provided. Without such a quantification of maternal hair nicotine, it is hard to base the authors’ conclusions on verbal or written answers.

The exposure to the secondhand smoking is this study is based on the validated questionnaires after conducting the pilot study with sample size of 40 women at the Obstetrics ward. This SHS questionnaires are available in the Supporting Information of the revised manuscript 

3. Were the pregnant women asked about the exposure to smoking prior to or after delivery? It is possible that the women looked for a reason for their preterm deliveries if they were asked after delivery.

This is a case-control study, so the interviews were conducted after delivery when the outcome (preterm birth, full-term birth) have already occurred. All women in the case and control group were asked the exposure to the secondhand smoking. The disadvantages of the case-control study were presented in the limitations of the study in the manuscript 

4. Did the authors compare the incidence of preterm birth between women with active smoking and women with secondhand exposure? This is very important because this has not been studied.

Nearly all Vietnamese women don’t smoke (1%), moreover the women who smokes were excluded from the present study. Therefore, we don’t have the data related to women with active smoking. All the pregnant women in this study were secondhand smoking 

5. Results: If the reviewer understands it clearly, 67.4% women with preterm birth revealed that they were exposed to secondhand smoke compared to 51% with term pregnancy. This means 51% women with term pregnancy also were exposed. Since 51% exposed women experienced term pregnancy, there must be a second confounding factor in women with preterm birth. Can the authors elaborate on this issue?

In a case-control study, as the outcome occurred (preterm of term), interviewers will ask about the past exposures (secondhand smoking), of course, the exposure was found in both groups, but the proportion of exposure was different between the two groups (67.5% in preterm vs. 51% in full-term group). From the difference we calculate the odds ratio. In this study OR1 for being preterm was 194/94 and OR2 for being term baby was 147/141, and crude OR will be: OR1/OR2= 194/94: 147/141=1.97. The secondhand smoking is the main predictor, not a confounding factor in this study. 

6. PLOS authors have the option to publish the peer review history of their article (what does this mean?). If published, this will include your full peer review and any attached files.

No

Do you want your identity to be public for this peer review? For information about this choice, including consent withdrawal, please see our Privacy Policy.

Reviewer #1: No

---

## [Decision Letter · Decision Letter 1]

24 Sep 2020

Preterm birth and secondhand smoking during pregnancy: a case–control study from Vietnam

PONE-D-20-09934R1

Dear Dr. Nguyen,

We’re pleased to inform you that your manuscript has been judged scientifically suitable for publication and will be formally accepted for publication once it meets all outstanding technical requirements.

Kind regards,

Cheryl S. Rosenfeld, DVM, PhD

Section Editor

PLOS ONE

Additional Editor Comments (optional):

Reviewers' comments:

Reviewer's Responses to Questions

**Comments to the Author**

1. If the authors have adequately addressed your comments raised in a previous round of review and you feel that this manuscript is now acceptable for publication, you may indicate that here to bypass the “Comments to the Author” section, enter your conflict of interest statement in the “Confidential to Editor” section, and submit your "Accept" recommendation.

Reviewer #2: All comments have been addressed

2. Is the manuscript technically sound, and do the data support the conclusions?

Reviewer #2: Yes

3. Has the statistical analysis been performed appropriately and rigorously? 

Reviewer #2: Yes

4. Have the authors made all data underlying the findings in their manuscript fully available?

Reviewer #2: Yes

5. Is the manuscript presented in an intelligible fashion and written in standard English?

Reviewer #2: Yes

6. Review Comments to the Author

Reviewer #2: The authors describe a retrospective case-control study investigating the potential influence of SHS on preterm birth rates in a cohort of Vietnamese women. This is a resubmission that addressed the comments of the previous review. The queries previously raised have been addressed.

A few issues could have been discussed in the discussion:

Does paternal cigarette somking affect sperm quality and DNA and therefore pregnancy outcome including risk of preterm birth? For instance Bodi et al 2019 Reproductive Toxicology 87:11-20 report the influence of paternal alcohol intake on fetal growth restriction and a comment regarding this potential confounder would have been relevant: preterm delivery may be attributable to the direct effect of paternal sperm changes from smoking on embryo quality and pregnancy duration rather than SHS.

It is not stated whether smokers within the household smoked in the house or outside.

Table 2 will benefit from a more detailed title - state that the table relates to preterm delivery before 37 weeks.

The limitations should have emphasised that the questionnaires were administered after delivery and therefore bias cannot be excluded.

7. PLOS authors have the option to publish the peer review history of their article (what does this mean?). If published, this will include your full peer review and any attached files.

Reviewer #2: No

---

## [Editor Report · Acceptance letter]

28 Sep 2020

PONE-D-20-09934R1 

Preterm birth and secondhand smoking during pregnancy: a case–control study from Vietnam 

Dear Dr. Rang:

I'm pleased to inform you that your manuscript has been deemed suitable for publication in PLOS ONE. Congratulations! Your manuscript is now with our production department. 

Kind regards, 

on behalf of

Dr. Cheryl S. Rosenfeld 

Section Editor

PLOS ONE